# An Assessment of the Applicability of Sustainability Measurement Tools to Resource-Based Economies of the Commonwealth of Independent States

**Tatyana Ponomarenko \*, Marina Nevskaya \* and Oksana Marinina \*** 

Faculty of Economics, Saint Petersburg Mining University, 199106 Saint Petersburg, Russia
\* Correspondence: stv@spmi.ru (T.P.); nevskaya_ma04@spmi.ru (M.N.); Marinina_OA@pers.spmi.ru (O.M.);
Tel.: +7-911-295-9667 (O.M.)

**Abstract:** The concept of sustainable development (SD) is aimed at ensuring public well-being for the present and future generations. Hundreds of methods have been proposed for assessing and comparing the sustainable development of countries and analyzing their contribution to the future of the world. When applied to resource-based economies (RBEs), assessment tools do not take into account the value and impact of mineral resources on SD indicators. The purpose of the study is to reveal the limitations of applying some tools by taking into consideration the specific features of RBEs. Research methods include a correlation analysis between gross national income (GNI) per capita and aggregated indices (the Sustainable Society Index (SSI), the Human Development Index (HDI), and the Environmental Performance Index (EPI)), a comparative analysis of these indices and mining companies' performance indicators. Object Eurasian RBEs were selected, but Norway was analyzed separately from the sample. The results of the study show that correlations between GNI per capita and SD indicators are heterogeneous. There is no statistically significant correlation between GNI per capita and SSI, a strong correlation with HDI, and a weak correlation with EPI. The EPI and SSI structures do not reflect the specific features of RBEs.

**Keywords:** sustainable development; assessment; indicators; resource-based economy; mineral resources

## 1. Introduction

Currently, the global community considers laying the groundwork for ensuring universal human well-being based on the principles of sustainability to be one of the highest priorities. Despite the fact that a lot of studies have been conducted over a long time [1–6], the concept of sustainable development (SD) has not yet been shaped into a scientific theory. SD is perceived as a fundamental principle or model for managing economic systems at macro, meso, and micro levels, as public administration ideology, or a new philosophy at the corporate level.

In this regard, the problem arises of sustainability assessment in terms of economic, environmental, and social indicators at the macro level (that of individual countries). An objective way of solving this problem is to carry out quantitative assessment based on a set of indicators or a combination of indicators in an integrated form. There are some well-known indicators which are used at the macro level to monitor economic, environmental, and social parameters, to develop strategies and programs for economic and social development, to identify indicators signaling problem areas, and to compare countries using different parameters.

The relevance of measuring sustainability in resource-based economies (RBEs) stems from the fact that their number and share in global GDP is growing. For example, at the end of the 20th century, 58 countries accounted for 18% of global GDP, whereas in 2011, 81 countries accounted for 26% of the

global economy [7]. RBEs are characterized by: a) a high share of income from mining operations in their GDP and export structure, b) being largely dependent on tax revenues from the mining sector, c) the depletion of mineral resources, d) huge amounts of waste accumulated in the mining sector, e) significant anthropogenic impacts on the environment, and f) massive investments in environmental protection measures. These characteristics give rise to the following research questions:

Is there a correlation between the economic indicator (The gross national income (GNI) per capita) and a number of integrated sustainability indicators for RBEs?

What is the impact of the Russian mining industry on the natural and social environment?

How are the specific features of RBEs (such as the depletion of resources, huge amounts of mining waste, anthropogenic impact, and environmental costs) reflected in the most popular sustainability assessment methods?

The concept of sustainable development is still debatable and has dozens of definitions, which makes it impossible to develop a single conceptual framework that could be used in further research [6]. Many researchers and institutions are trying to expand the famous definition given by the Brundtland Commission by adding to it the specifics of either particular countries or activities [8]. For example, the two definitions given by the Brundtland Commission are as follows: the Sustainable Society Foundation (SSF) has extended the definition of Brundtland with a third sentence, as follows: "A sustainable society is a society that (a) meets the needs of the present generation; (b) does not compromise the ability of future generations to meet their own needs; and (c) in which each human being has the opportunity to develop itself in freedom within a well-balanced society and in harmony with its surroundings".[9]. As the components of sustainable development are complex, interconnected, interdependent, ambiguous, and difficult to assess, sustainable development can be called a global goal that is difficult to reach and the progress towards which is very slow. In 2002, [10] it was concluded that global economic, environmental, and social problems had not been solved. This was the beginning of a new wave of interest in this topic demonstrated by both developed and developing countries.

So far, three main approaches to building a theory of sustainable development have been developed: the ecosystem approach, the anthropocentric approach, and the triune approach. An analysis of these approaches has shown that while none of them is universally recognized, the triune approach is becoming predominant in SD research. This is a result of the activities of international organizations that are involved in monitoring and forecasting change in both the global economy and countries with different levels of economic development.

In its modern interpretation, the ecosystem approach is a viewpoint which is based on the priority of the environmental component and argues that sustainable development at both global and national levels means that the economy, population, and people's needs should grow within the resource and environmental limits of the planet or an individual country. This approach is based on the principles of environmental economics and the concept of strong sustainability [1,11–13].

An alternative to the ecosystem concept is the anthropocentric approach, which implies that economic growth is fostered by the development of human capital and technological advances, which weaken the impact of limited natural resources on the economy. This concept is based on the principles of neoclassical economics and the concept of weak sustainability. The main idea is that economic growth is independent of its impact on the environment [14]. Research and development (R&D) become an integral part of sustainable development strategies [15].

The concept of weak sustainability is based on the requirements specified by people as to the quality of the environment in order to satisfy their needs. However, this approach does not challenge the need for the coordinated development of the economy and environment, the rational use of resources and natural resource restoration, biodiversity conservation, and environmental safety [16]. At the same time, there is a point of view which states that there is no such thing as weak stability, since the current economic system can be characterized as unstable and weak stability will not result in any form of sustainable future [4]. The main arguments against the concept of weak sustainability are that innovative solutions are not always available [17], and that the use of environmentally friendly

technologies often requires much more energy and resources if the whole production process is analyzed [4].

The anthropocentric approach to sustainable development laid the foundation for the concept of a green economy, which also has global and national (sectoral, territorial) components. On a global scale, UNEP experts define a green economy as being economic activities that improve human well-being and promote social justice while significantly reducing environmental risks and preventing environmental degradation [18]. On a national scale, the concept of a green economy is mainly applicable to developed European countries that are rich in innovations, investment, and human resources but lack natural resources, especially mineral ones, which they have to import.

In resource-based countries (those rich in natural resources but lacking innovations), including Russia, the concept of a green economy and its derivatives (green energy, green mining) are also considered to be components of sustainable development. However, they are mainly regarded as tools for improving economic performance and upgrading production facilities in terms of their environmental characteristics. As the primary industries of the Russian economy, including the mining industry, lag behind in upgrading their facilities and making them greener [19], mineral resources can be greatly underestimated. Moreover, there is a danger that environmentally unfriendly facilities will be moved to countries with a high mineral resource potential.

One of the main directions in the development of green economy ideas is the concept of a circular economy, which states that it is of bigger priority to rearrange material flows than improve production facilities [20]. Considering the fact that this concept rests on the ideas of resource cycles and industrial metabolism, we believe that it can be considered to be a compromise between the concepts of strong and weak sustainability. The economic transformation of the "take, make, waste" principle into the "take, make, reuse" principle corresponds to weak sustainability. Compliance with the principles of resource efficiency and zero waste corresponds to strong sustainability [21].

SD relying on circular economy ideas has in its foundation the following five principles (5 Rs):

-    reducing energy and resource consumption,
-    replacing non-renewable resources with renewable ones,
-    recovering necessary components from waste that has been recycled,
-    recycling waste,
-    reusing products.

From an economic point of view, recycling can increase the material value through efficient resource processing, providing new opportunities for innovative companies, and having a positive impact on society and the environment [22]. The circular economy implies that there is enhanced control over the supply of natural resources which relies on renewable resources in order to protect and enhance natural capital. It also means that it is necessary to optimize consumption processes through reusing and recycling products. Moreover, it is aimed at identifying the sources of negative impact from production processes on the environment and preventing negative consequences in order to improve the efficiency of economic and environmental systems [21].

The ecosystem and anthropocentric approaches define the place and role that people and society play in sustainable development in different ways. In the strong sustainability model, society is seen as a source of needs that should be reasonably limited, while the weak sustainability model defines it as a source of human resources that need to be transformed into human capital for innovative development. The inclusion of people as a particular subject in the framework of sustainable development is embedded in the triune approach.

In the context of this approach, there are three main goals of sustainable development: environmental integrity, eco-efficiency, and environmental justice. The economic component of the triune approach is based on the Hicks-Lindahl concept of the maximum income stream, which can be produced provided that the total capital used for its production is maintained [23]. In this concept, it is not taken into account which kind of capital (human, natural, or material) needs to be maintained

or increased. Thus, the two previously discussed concepts of strong and weak sustainability are taken into account.

The social component of the triune approach implies that cultural and social systems should be preserved, with material wealth justly distributed. The social component of SD is understood as development aimed at improving well-being, achieving greater social justice, and improving the quality of human and social capital to make it meet the principles of reproducibility, balance, and involvement [24].

The environmental component implies that biological systems should be preserved and their stability over time should be maintained. It can be said that the triune approach combines the concepts of strong and weak sustainability, complementing them with a social element (Figure 1).

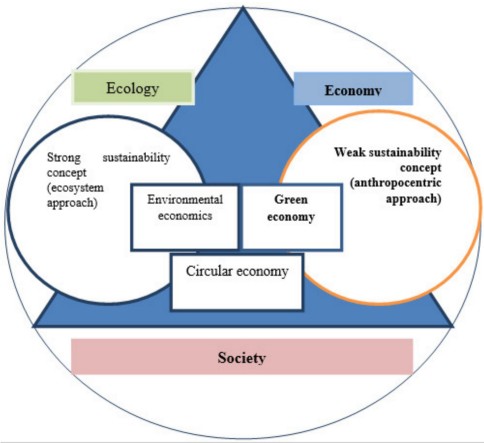

**Figure 1.** Relationships between sustainable development concepts within the triune concept. Source: compiled by the authors.

The triangle is a three-pronged approach with equivalent environmental, economic, and social goals for SD.

Circles are the two main concepts of sustainable development (strong sustainability and weak sustainability).

Rectangles are the main economic tools and principles that are characteristic of the concepts of strong sustainability (tools and principles of environmental Economics), weak sustainability (tools and principles of "green" economy), and the triune approach (tools and principles of circular economy).

It should be emphasized that the anthropocentric approach to SD is more characteristic of developed countries with high levels of development and income. Such countries are characterized by a big share of human and social capital (up to 80%), a small share of produced capital (less than 20%), and a very small share of natural capital (up to 5%), (Figure 2).

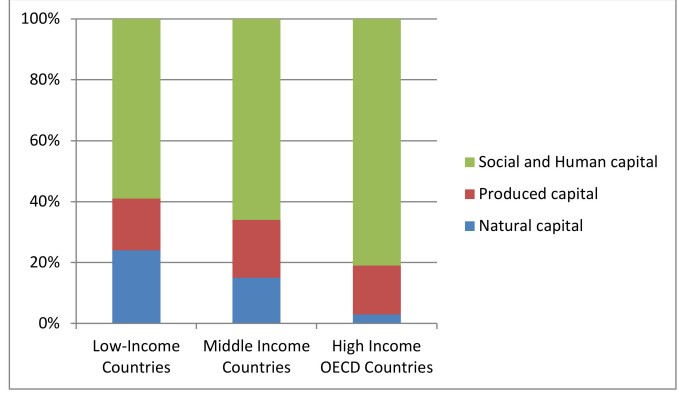

**Figure 2.** Capital structure by different levels of economic development [25].

The ecosystem approach based on the principles of natural resource restoration and environmental damage mitigation can be seen in resource-based countries. It should be noted that these countries are very different in terms of their levels of economic and institutional development. So far, no clear differentiation criteria have been developed that take into account the country's industry specialization which determines its place in the global economy and the specific conditions for its sustainable development.

Certain conflicts between the ecosystem and anthropocentric approaches have been identified in terms of how important natural resources, including mineral ones, are considered to be for countries' social and economic development. This is due to the fact that the role that mineral resources play at a macroeconomic level varies a lot. In some developed countries, the share of the mining industry in the country's GDP can amount to 10% (for example, in Sweden), whereas in resource-based countries, it can exceed 50% [26]. Resource-based countries have different quantities of mineral reserves. They vary in how developed their mining and mineral processing industries are and how big their impact on the environment is. Also, they differ in terms of multiplicative effects witnessed in related industries. As a result, they differ in their economic, environmental, and social indicators and are at different levels of sustainable development.

Among resource-based countries, the leading ones are developing countries with economic development levels that are both above the average and low. Most of them are former colonies and former Soviet countries, including Russia. In the case of these countries, which traditionally specialize in producing mineral and energy resources, a sustainability assessment is impossible without taking into account the specific impact of the natural resources sector on social and economic development and how this impact can be measured. However, as the issue of how to define sustainable development has not yet been solved, no framework for measuring it at the macro level has been created.

Studies by S. Kuznets in assessing national product and income as indicators of change marked the start of using macroeconomic indicators in assessing countries' levels of economic development. In the 1970s, the first attempts were made to switch from national product and national income to more complex indicators which would make it possible to take into account the influence of non-market forces on economic development at the macro level. It should be noted that S. Kuznets himself drew attention to the fact that it is possible to expand the range of assessment tools to include those that can take into account non-market activities and emphasized that assessments depend on the social values of time and place, the nature of family, industrial, and governmental organization, and the problems to which the figures are designed to apply [27]. Thus, in his approach, S. Kuznets emphasized the connection between social well-being and stakeholders' interests.

Sustainability measurement tools include about 500 indices. It should be noted that they hardly take into account the specialization of the economy or company. However, resource-based countries have several features which determine how a sustainable development concept will be designed and implemented:

-　　the exploitation of natural resources and the associated depletion of non-renewable mineral resources;
-　　the accumulation of mining waste and the human impact which reduces the efficiency of ecosystem services;
-　　the need to bear additional environmental costs (including both investment and operational costs).

In their studies, a number of authors have analyzed the most common and relevant indices [28–36]. However, there have been few attempts to systematize them. There is no single classification of indicators, and neither is there a universal method for assessing sustainability in an objective way. Such a situation, on the one hand, makes it more difficult to choose assessment methods and indicators. On the other hand, it indicates that these methods are constantly being improved (some indicators and indices become replaced with others, the number of assessment indicators is changing, and more countries are being included) but, unfortunately, not in keeping with real-life social, economic, and environmental processes.

All the assessment methods that have been developed so far can be classified on the basis of several characteristics:

- How the method is developed: systems of indices and indicators; aggregated indicators.
- What parameters are assessed: environmental; economic; social; social and environmental; environmental and economic.
- How information is accumulated: through open-access international databases (secondary); through the SNA and using statistical data (secondary); polls (secondary).
- How different indicators are combined: using ranking systems; using weighting factors.
- What methods are used: calculation; expert-based evaluation.
- What countries are covered: all countries; OECD countries; countries by continents; countries grouped by other parameters.
- What issues are covered [37]: environmental rankings; rankings reflecting how individual countries impact the environment on a global scale; rankings including both environmental and economic parameters; social development rankings; rankings based on sustainable development indicators; rankings reflecting progress in the green economy; ratings reflecting the quality of life and including the environmental component; other rankings including the environmental component.
- According to Hezri and Dovers [38] (p. 87), the main approaches to developing sustainability indicators are as follows: (1) extended national accounts, (2) bio-physical accounts, (3) weighted indices, (4) eco-efficiency and dematerialization approaches, and (5) indicator sets.

Among the systems of indicators (indices), the most common are World Development Indicators [39], the OECD system of indicators [40], and the system of indicators by the UN Commission on Sustainable Development (UN CSD). Their advantage is that they are focused on the SDGs. However, they rely on a lot of input data and cannot be used to compare individual countries. This is why it is difficult to apply them to sustainability assessment in real life.

The most common aggregated SD indicators include the Adjusted Net Savings (ANS) [41], the Genuine Progress Indicator (GPI), the Ecological Footprint, the Environmental Performance Index (EPI), the Legatum Prosperity Index, the Human Development Index (HDI), the Living Planet Index (LPI), the Sustainable Society Index (SSI), and others.

Sustainability assessment indicators should meet the following requirements:

Indicators should not be in conflict with the requirements of the SDGs and the indicator system developed by the UN CSD.

They should be universal to be applied at different levels (that of individual countries, industries, or companies) and they should adequately reflect the situation and processes.

Indicators should be obtained or calculated using relevant and reliable sources of information (statistical ones) that are publicly available.

Indicators should take into account the economic, environmental, and social aspects of SD and how they are interconnected.

## 2. Materials and Methods

The statistical correlation between the key economic indicator that characterizes the level of economic development of a country (GNI per capita) and indices that characterize the three aspects of SD was revealed as follows.

### 2.1. The Object of the Study

As an object of the study, Commonwealth of Independent States (CIS) resource-based countries were selected. The main criteria for studying RBEs were proposed by the IMF, the Natural Resource Governance Institute (NRGI) [42], and several researchers. Studies devoted to resource-based countries arose due to the fact that there was an ambiguous attitude to natural resources as, on the one hand, an essential asset for improving the economy and fostering economic growth, and, on the other hand,

an asset resulting in the so-called resource curse [43,44]. In 2005, the latter concept was discussed by the UN and it was shown that an important role in this issue is played by institutions and how developed they are. Australia, Canada, and Scandinavian countries were used as examples. This was when a resource-based economy was given its definition which states that in such an economy, natural resources account for more than 10% of the country's GDP and 40% of exports. The IMF suggests that as a benchmark, the share of natural resources in total exports should be at least 25% [45]. An important characteristic of RBEs is that they are highly sensitive to external changes due to the volatility of prices for different types of raw materials [46]. In this study, two criteria (GDP and export shares) were selected as the main ones [47].

As objects, we select countries that correspond to two conditions at the same time:

(1) CIS countries with transition economies:

Source: https://www.imf.org/external/np/exr/ib/2000/110300.htm.

(2) countries with economies that can be characterized as resource-based.

Thus, the object of the study is the following six countries: Russia, Kazakhstan, Kyrgyzstan, Uzbekistan, Turkmenistan, and Azerbaijan.

## 2.2. Selection of Assessment Indicators

Among the indices that characterize SD at the macro level, SSI, HDI, and EPI were selected. The choice was made using Kostanza's classification by the method of measuring progress in SD [29] using each group of indicators.

As an integral indicator covering economic, social, and environmental aspects, the Sustainable Society Index (SSI), which has been calculated by the Sustainable Society Foundation (SSF) since 2006 once every two years, was selected. This index connects economic performance with social and environmental well-being. It has scores ranging from zero (minimum SD) to ten (maximum SD) and takes into account 21 indicators in three areas: human well-being, environmental well-being, and economic well-being [48]. The methodology for calculating the index varies depending on the set of indicators and their relative shares [49]. The Sustainable Society Index (SSI) aims to be a comprehensive and quantitative method to measure and monitor the health of coupled human-environmental systems at a national level worldwide. The SSI framework departs from a purely protectionist approach that would aim to maintain natural systems with minimal human impact [9].

As a social indicator, the Human Development Index (HDI) was selected, which is used to compare standards of living in different countries.

HDI is calculated as follows [50]:

$$HDI = \sqrt[3]{LEI \cdot EI \cdot II} \tag{1}$$

where LEI is the life expectancy index, EI is the education index, and II is the income index.

The index has a 30-year-long history; since 1990, HDI indicators for different countries have been published annually in the Human Development Reports by the United Nations Development Programme (UNDP).

As an environmental indicator, the Environmental Performance Index (EPI) was selected, which is used as a tool for quantitative assessment and comparative analysis to analyze the environmental situation and whether countries' environmental policies are effective. The purpose of the index is to reduce environmental impact and, as a result, the negative impact on human health, and to foster the vitality of environmental systems and the sustainable management of natural resources.

The EPI system ranks countries based on their performance in several categories, which are combined into two groups: ecosystem vitality and environmental health [51]. EPI rankings are published every two years and are calculated using the methodology developed by the Yale Center for Environmental Law and Policy together with a group of independent international experts. According to the developers, the higher the score, the more the country cares about the environment.

Based on the 2018 methodology, the index measures the country's performance using 24 indicators across 10 issue categories, which reflect various environmental aspects including those of ecosystem vitality [52]. They include protecting biodiversity, climate, and public health, the scope of economic activity and its impact on the environment, as well as the effectiveness of environmental policies [51]. The calculation method, the number of indicators, and the number of countries vary in different years. For example, the number of indicators was 22 in 2012, 19 in 2014, and 24 in 2018. The weighting factors for different groups and categories can also change. Following the adoption of the Global SDGs in 2015, the EPI indicators have been aligned with the goals.

The indicators described above were selected due to several reasons [51]:

- the most important factor for health is clean air. A study conducted in 2016 by the Institute for Health, Metrics and Evaluation showed that 2/3 of all diseases and deaths are connected with air quality;
- the quantity and quality of biomass (both in the sea and on land) are of great importance, which corresponds to the SDGs;
- many countries have improved air quality by reducing $CO_2$, $NO$, methane, and black carbon emissions, which is in line with the 2015 Paris Climate Agreement;
- over 20 years of research, experience has been accumulated which shows that when developing indices, two opposite patterns should be taken into account: environmental health, which improves with economic growth and development, and ecosystem vitality, which worsens with industrialization and the expansion of economic activity.

In view of this, countries with a growing and developing resource-based economy which depends on the mining industry can obviously rank very low.

*2.3. Relevance Score*

An analysis of the correlation between the selected assessment indices and GNI per capita in RBEs was performed. The per capita indicator was selected due to the need to ensure that income indicators for different RBEs can be compared. For the purpose of comparability, GDP values were adjusted and presented as normalized values. Normalization was performed for the entire set of countries in the sample. Then, we get a range of values from 0 to 1 on normalized GDP scales. The indexes characterize the country's place in the global rankings and estimate normalized scoring. The method of regression and correlation analysis makes it possible to test the hypothesis of the study which states that there should be a rather strong correlation between GNI per capita, human development, environmental indicators, and social sustainability since the country's income provides financial support for social and environmental programs. If this correlation is positive, there is no evidence of a "resource curse" in the macroeconomic sense, mineral resources in RBEs have a positive impact on the development of the economy, the environment, and society. If this correlation is negative, then according to the institutional interpretation of the "resource curse" [53], RBEs have an undeveloped institutional environment.

The method of morphological analysis was used to analyze the SD indices (SSI, HDI, EPI, Adjusted net savings (ANS), and GPI) and reveal how their structures reflect the specific features of resource-based countries. The initial data were obtained from official reports by the UN and the WB [54–59].

Using Russia as an example, changes in the indicators showing waste accumulation, investment and operational costs associated with environmental protection, and investment patterns in the mining sector were analyzed.

## 3. Results

The initial data were obtained from official reports by the UN and the WB (Table 1).

To find correct correlations between the indices and GNI per capita values, the latter were normalized and the indices were found.

**Table 1.** Gross national income (GNI) per capita in Commonwealth of Independent States (CIS) resource-based economies (RBEs), USD (thousands).

| Country | 2010 | 2011 | 2012 | 2013 | 2014 | 2015 | 2016 | 2017 | 2018 |
|---|---|---|---|---|---|---|---|---|---|
| Kyrgyzstan | 0.78 | 0.87 | 0.83 | 0.9 | 0.99 | 1.21 | 1.25 | 1.17 | 1.1 |
| Uzbekistan | 0.91 | 1.1 | 1.28 | 1.51 | 1.72 | 1.88 | 2.09 | 2.16 | 2.22 |
| Turkmenistan | 2.84 | 3.42 | 3.79 | 4.8 | 5.41 | 6.88 | 8.02 | 7.38 | 6.67 |
| Azerbaijan | 3.83 | 4.84 | 5.33 | 5.29 | 6.22 | 7.35 | 7.6 | 6.56 | 4.76 |
| Kazakhstan | 6.16 | 6.92 | 7.58 | 8.26 | 9.78 | 11.55 | 11.85 | 11.39 | 8.81 |
| Russia | 9.66 | 9.34 | 9.9 | 10.65 | 12.7 | 13.85 | 13.22 | 11.72 | 9.72 |

Normalization was performed for all the six countries in the sample for each year.

The model for calculating GNI indices has the following form ($I_{GNI}$):

$$I_{GNI} = \frac{(GNI_f - GNI_{min})}{(GNI_{max} - GNI_{min})} \qquad (2)$$

where $GNI_f$, $GNI_{min}$, $GNI_{max}$ are factual, minimum, and maximum values of GNI per capita (thousand $/person).

The results of calculating GNI per capita are shown in Table 2.

**Table 2.** Normalized GNI per capita in CIS RBEs.

| Country | 2010 | 2011 | 2012 | 2013 | 2014 | 2015 | 2016 | 2017 | 2018 |
|---|---|---|---|---|---|---|---|---|---|
| Kyrgyzstan | 0.000 | 0.000 | 0.000 | 0.000 | 0.000 | 0.000 | 0.000 | 0.000 | 0.000 |
| Uzbekistan | 0.015 | 0.027 | 0.050 | 0.063 | 0.062 | 0.053 | 0.070 | 0.094 | 0.130 |
| Turkmenistan | 0.232 | 0.301 | 0.326 | 0.400 | 0.377 | 0.449 | 0.566 | 0.589 | 0.646 |
| Azerbaijan | 0.343 | 0.469 | 0.496 | 0.450 | 0.447 | 0.486 | 0.530 | 0.511 | 0.425 |
| Kazakhstan | 0.606 | 0.714 | 0.744 | 0.755 | 0.751 | 0.818 | 0.886 | 0.969 | 0.894 |
| Russia | 1.000 | 1.000 | 1.000 | 1.000 | 1.000 | 1.000 | 1.000 | 1.000 | 1.000 |

HDI and SSI in CIS RBEs are shown in Tables 3 and 4.

**Table 3.** HDI in CIS RBEs.

| Country | 2010 | 2011 | 2012 | 2013 | 2014 | 2015 | 2016 | 2017 | 2018 |
|---|---|---|---|---|---|---|---|---|---|
| Azerbaijan | 0.732 | 0.731 | 0.736 | 0.741 | 0.746 | 0.749 | 0.749 | 0.752 | 0.754 |
| Kazakhstan | 0.764 | 0.772 | 0.782 | 0.791 | 0.798 | 0.809 | 0.808 | 0.813 | 0.817 |
| Kyrgyzstan | 0.636 | 0.639 | 0.649 | 0.658 | 0.663 | 0.666 | 0.669 | 0.671 | 0.674 |
| Russia | 0.78 | 0.789 | 0.797 | 0.803 | 0.807 | 0.813 | 0.817 | 0.822 | 0.824 |
| Turkmenistan | 0.673 | 0.68 | 0.686 | 0.691 | 0.696 | 0.701 | 0.706 | 0.708 | 0.71 |
| Uzbekistan | 0.665 | 0.672 | 0.681 | 0.688 | 0.693 | 0.696 | 0.701 | 0.707 | 0.71 |

**Table 4.** SSI components in CIS RBEs.

| Country | 2010 | | | 2012 | | | 2014 | | | 2016 | | |
|---|---|---|---|---|---|---|---|---|---|---|---|---|
| | Human Well-Being Index | Environmental Well-Being Index | Economic Well-Being Index | Human Well-Being Index | Environmental Well-Being Index | Economic Well-Being Index | Human Well-Being Index | Environmental Well-Being Index | Economic Well-Being Index | Human Well-Being Index | Environmental Well-Being Index | Economic Well-Being Index |
| Azerbaijan | 6.9 | 4.9 | 5.6 | 6.9 | 5.5 | 5.7 | 7.1 | 5.1 | 5.9 | 7.3 | 3.9 | 5.7 |
| Kazakhstan | 7.4 | 2.1 | 4.2 | 7.4 | 2.8 | 4.2 | 7.5 | 2.5 | 4.6 | 7.6 | 2.7 | 5.3 |
| Kyrgyzstan | 6.9 | 4.9 | 3.1 | 7 | 6.2 | 3.5 | 7.1 | 4.9 | 3.2 | 7 | 4.9 | 2.2 |
| Russia | 6.9 | 2.3 | 5.2 | 6.8 | 2.5 | 5.1 | 6.8 | 2.3 | 5.4 | 6.9 | 2.5 | 5.5 |
| Turkmenistan | 5.7 | 1.5 | 4.6 | 5.6 | 1.7 | 4.6 | 5.8 | 1.8 | 4.9 | 5.8 | 1.7 | 4.9 |
| Uzbekistan | 6.4 | 5.0 | 3.7 | 6.2 | 5.3 | 3.8 | 6.2 | 5.1 | 3.9 | 6.6 | 5.1 | 4 |

The average SSI values were calculated as the arithmetic mean due to the absence of this value in the initial data (Table 5).

**Table 5.** Average SSI in RBEs.

| Country | Average SSI Values | | | |
| --- | --- | --- | --- | --- |
| | **2010** | **2012** | **2014** | **2016** |
| Azerbaijan | 5.78 | 6.03 | 6.03 | 5.63 |
| Kazakhstan | 4.58 | 4.80 | 4.87 | 5.20 |
| Kyrgyzstan | 5.00 | 5.57 | 5.07 | 4.70 |
| Russia | 4.78 | 4.80 | 4.83 | 4.97 |
| Turkmenistan | 3.92 | 3.97 | 4.17 | 4.13 |
| Uzbekistan | 5.04 | 5.10 | 5.07 | 5.23 |

EPI in CIS RBEs are shown in Table 6.

**Table 6.** EPI in CIS RBEs.

| Country | EPI | | | | Average SSI | | | |
| --- | --- | --- | --- | --- | --- | --- | --- | --- |
| | **2010** | **2012** | **2014** | **2016** | **2010** | **2012** | **2014** | **2016** |
| Azerbaijan | 59.1 | 43.11 | 55.47 | 83.78 | 5.78 | 6.03 | 6.03 | 5.63 |
| Kazakhstan | 57.3 | 32.94 | 51.07 | 73.29 | 4.58 | 4.80 | 4.87 | 5.20 |
| Kyrgyzstan | 59.7 | 46.33 | 40.63 | 73.13 | 5.00 | 5.57 | 5.07 | 4.70 |
| Russia | 61.2 | 45.43 | 53.45 | 83.52 | 4.78 | 4.80 | 4.83 | 4.97 |
| Turkmenistan | 38.4 | 31.75 | – | 70.24 | 3.92 | 3.97 | 4.17 | 4.13 |
| Uzbekistan | 42.3 | 32.24 | 43.23 | 63.67 | 5.04 | 5.10 | 5.07 | 5.23 |

Figures 3–5 show the correlations between 1) GNI per capita and HDI, 2) GNI per capita and SSI, and 3) GNI per capita and EPI.

The sample of RBEs is characterized by a strong correlation between GNI per capita and HDI, which is almost linear. This may be due to the fact that GNI per capita is included in the HDI calculation model. It is very possible that for these countries the contribution of GNI per capita to the HDI model is more important than that of other components (such as the education index and the life expectancy index). This is why a lower GNI value correlates with a lower HDI value.

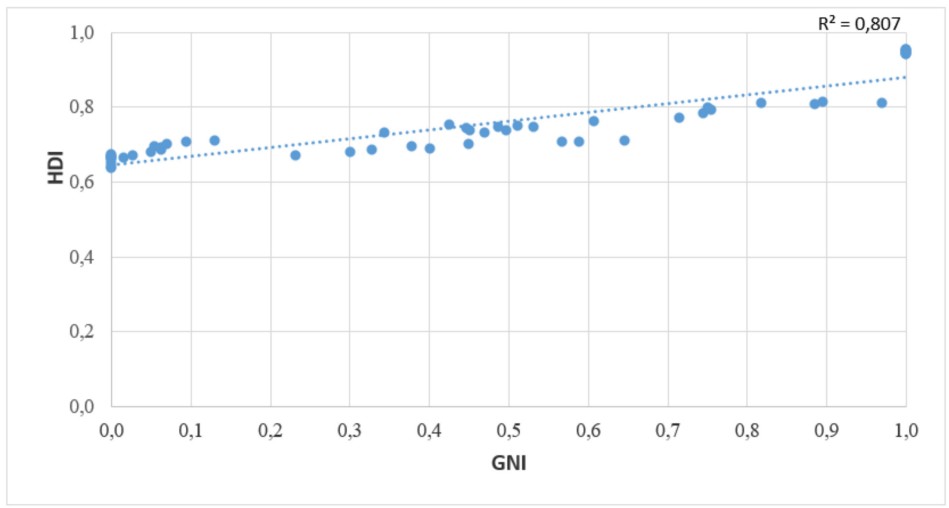

**Figure 3.** The correlation between GNI and HDI in CIS RBEs.

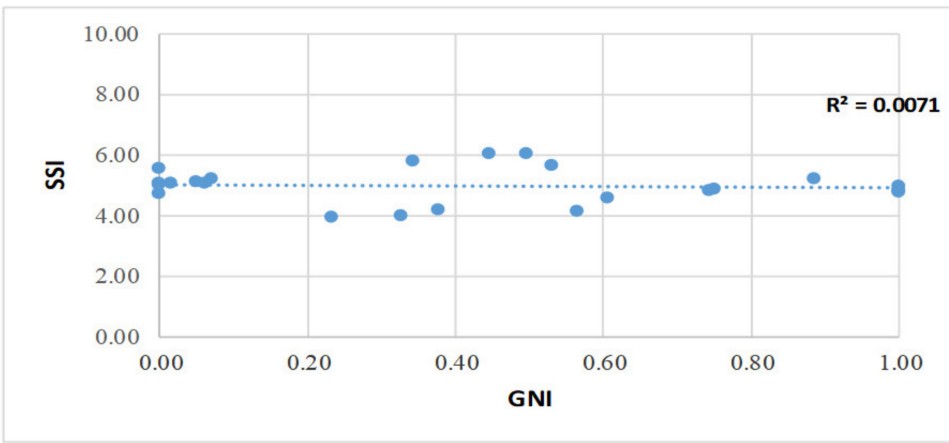

**Figure 4.** The correlation between GNI and SSI in CIS RBEs.

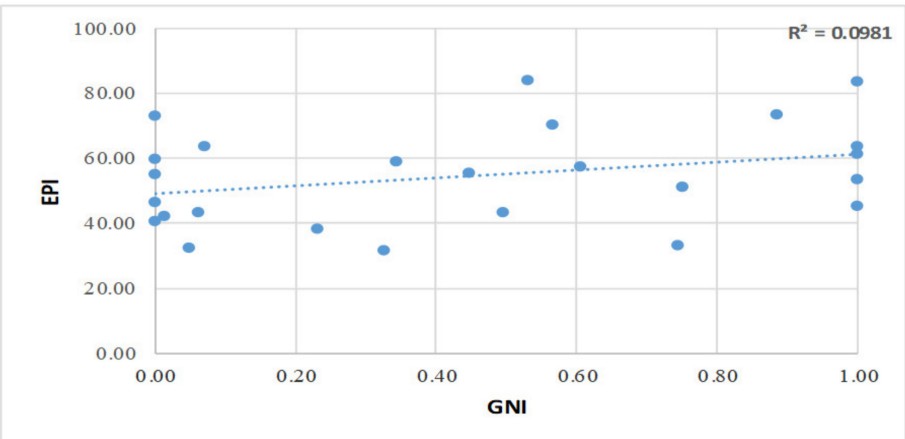

**Figure 5.** The correlation between GNI and EPI in CIS RBEs.

The results of finding the strength of the correlation between GNI per capita and SSI and that between GNI per capita and the SSI components are heterogeneous. For the group of countries, there is almost no correlation with SSI. SSI includes 21 indicators that can change their values in different directions, which leads to a complete lack of any correlation. It should be noted that SSI values for previous years are recalculated as the methodology changes, which ensures that the results are comparable.

The values of the correlation ratios between GNI per capita and EPI show a high variability with a low degree of correlation. EPI is based on 24 indicators characterizing completely different aspects of environmental health and ecosystem vitality. Such a number of indicators and the fact that changes are frequently made to the methodology result in a rather weak correlation.

Based on the statistical data for Russia, an analysis of the environmental impact of the mining sector was carried out, which showed that it strongly influences all elements of the natural environment including the atmosphere, biosphere, hydrosphere, and lithosphere. In terms of impact complexity, this sector ranks first among all other sectors of the economy. However, individual elements of the environment are impacted in different ways. According to the report titled Basic Environmental Protection Indicators which was issued in 2019 by the Russian Federal State Statistics Service (Rosstat) [60], the mining sector accounted for 10.8% of the total amount of greenhouse gas emissions (whereas the share of the energy sector was 78.8%). The volume of effluent did not exceed 6% of the total amount. At the same time, the mining sector is a leader in air pollutant emissions (with a share of more than 28%) and the generation of waste. A bar chart reflecting the situation with waste generation in Russia is shown in Figure 6.

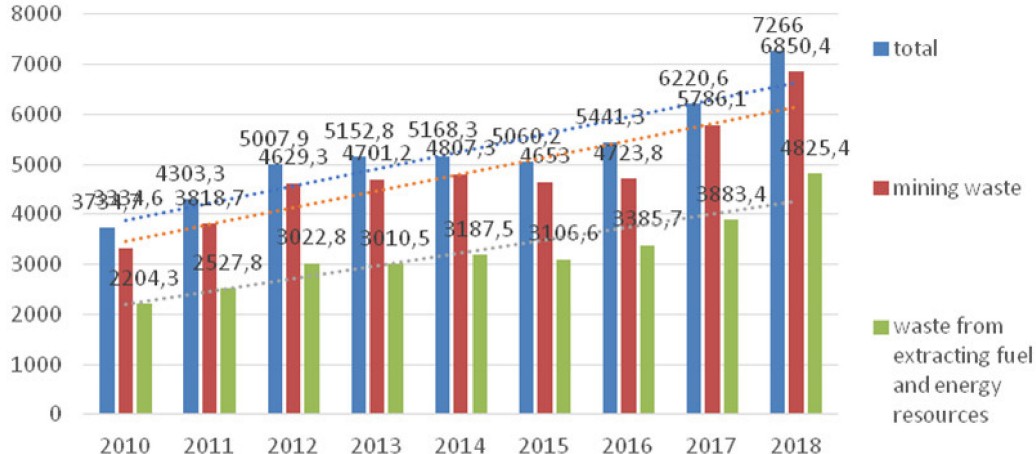

**Figure 6.** Waste generation in Russia, mln tons (compiled by the authors according to Rosstat data on environmental issues [60]).

More than 90% of waste generated in Russia is accounted for by the mining sector, of which about 70% is waste associated with the extraction of fuel and energy resources. More than half of the waste produced in the mining sector is recycled (Figure 7), and the rest goes to spoil tips or tailings dams, which can act as secondary sources of pollution.

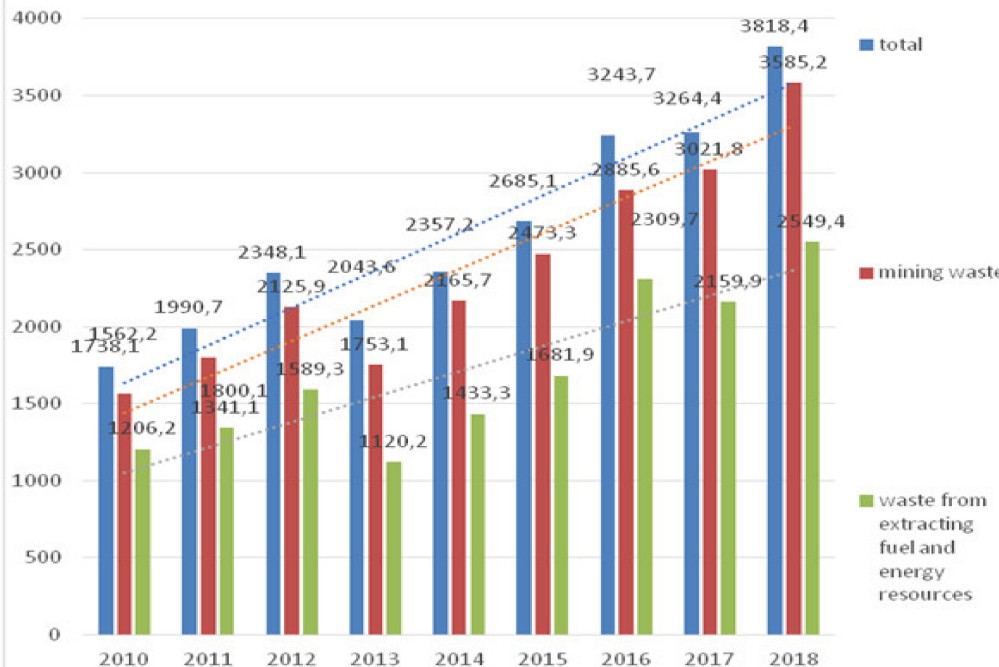

**Figure 7.** Waste recycling in the Russian Federation, mln tons (compiled by the authors according to Rosstat data on environmental issues [60]).

The complex nature of the environmental impact results in a high share of investment and operational costs associated with environmental protection in the mining sector. In 2017, out of the total amount of investment in environmental protection, which was 154.0 billion rubles, the mining sector accounted for more than 30%. In 2018, this share slightly decreased and amounted to 23.1% of 157.6 billion rubles [60]. Over the last 8 to 10 years, there has been an increase in both investment and operating costs associated with environmental protection in Russia (Figures 8–11).

Compared to 2012, there has been a 33% growth in operational costs associated with environmental protection. The main items are effluent disposal and treatment, waste management, air quality

protection, and climate change prevention. Over a ten-year period, the amount of investment in fixed assets associated with environmental protection almost doubled, and the main expenses were aimed at protecting the atmosphere and water resources. The share of investment in fixed assets in the GDP of the Russian Federation in 2017 and 2018 was 0.16% and 0.15%, respectively, and the share of operational costs in GDP did not exceed 0.3%, which is clearly not enough if we compare it with that demonstrated by developed countries. For example, the share of environmental control costs is approximately 1.47% of GDP in the United States, 1.5% of GDP in Germany, and 1.25% of GDP in Japan [61].

More than 50% of investment was aimed at protecting the atmosphere; a little more than 20% of investment was allocated for protecting water resources; a little more than 10% of investment was invested in waste management; less than 10% of investment was invested in land conservation. We believe that this can be explained by the fact that Russian legislation concerning mineral resource production considers mining waste to be a potential source of raw materials, and the activities associated with its use are qualified as activities related to the use of mineral resources.

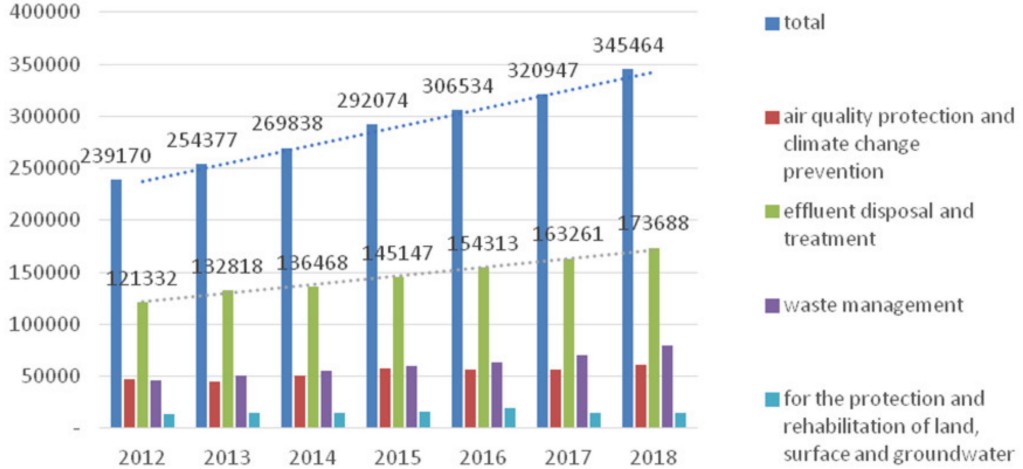

**Figure 8.** Operational costs associated with environmental protection in Russia, mln RUB (compiled by the authors according to Rosstat data on environmental issues [60]).

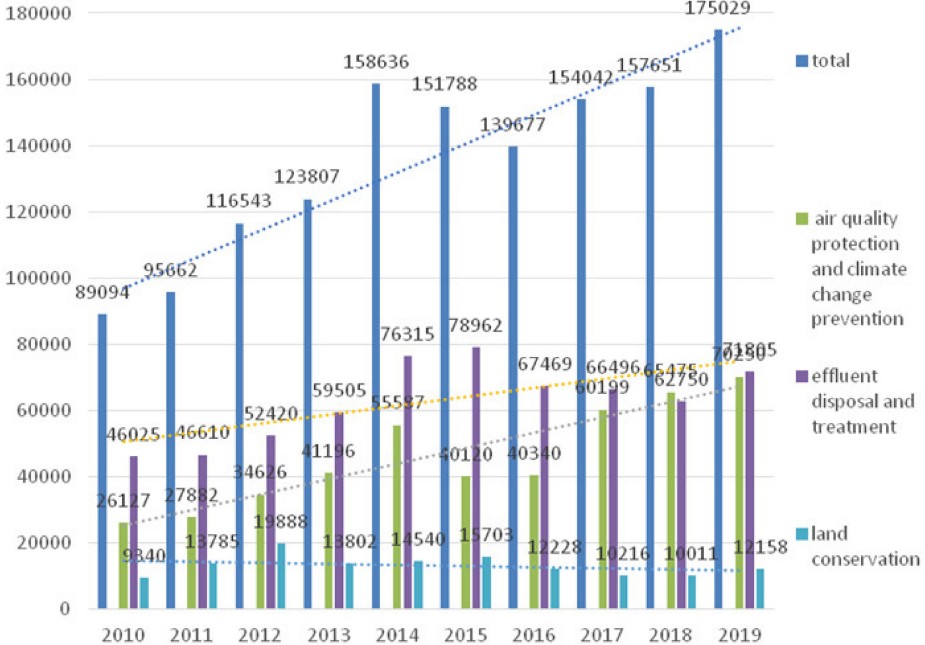

**Figure 9.** Investments in fixed assets associated with environmental protection, mln RUB (compiled by the authors according to Rosstat data on environmental issues [60]).

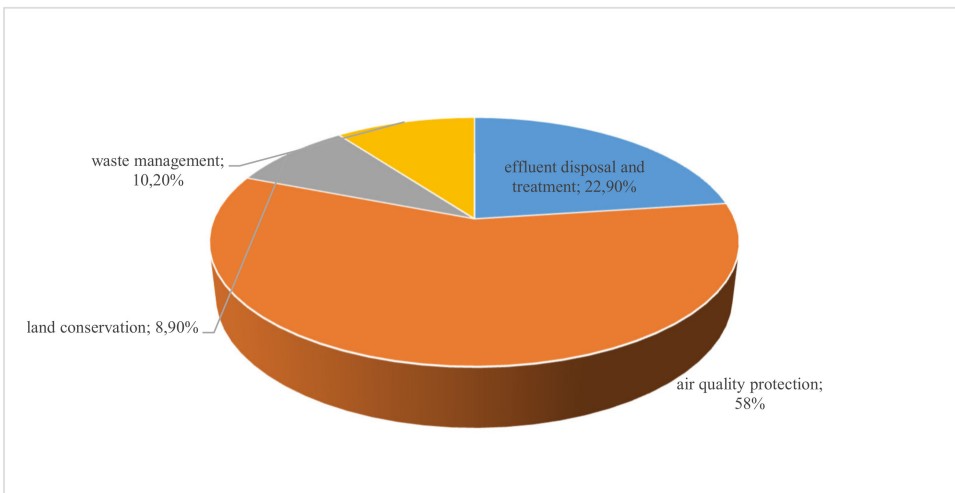

**Figure 10.** The structure of investment costs associated with environmental protection in the Russian mining sector in 2018 (compiled by the authors according to Rosstat data on environmental issues [60]).

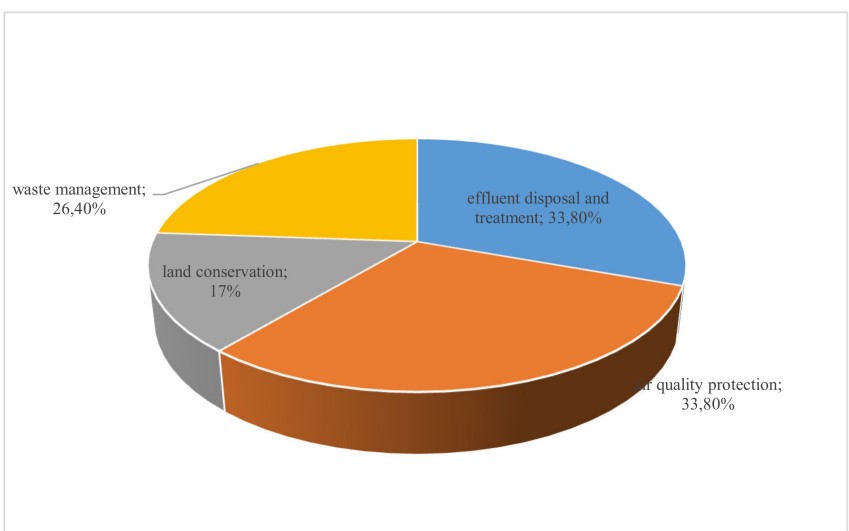

**Figure 11.** The structure of operational costs associated with environmental protection in the Russian mining sector in 2018 (compiled by the authors according to Rosstat data on environmental issues [60]).

## 4. Discussion

Results obtained by other researchers [30] who tested correlations between the SSI components for 151 countries over the years 2006, 2008, 2010, and 2012 prove a strong positive correlation between the three indices and a strong negative correlation between the human well-being index and the environmental well-being index.

An analysis of the SSI and EPI structures showed that they contain some indicators reflecting such aspects as agriculture, biodiversity, climate, water resources and air quality, energy use, water, and energy conservation. Issue-specific indicators reflect negative impact on the environment by categories: heavy metals, air quality and pollution, climate, and energy [61–63] In issue-specific indicators, greenhouse gases are separately analyzed as a negative factor affecting the climate. Russia, Kazakhstan, and Turkmenistan have been characterized by low scores over a long time [64,65] Russia takes 4th place in the world [66] after the USA, China, and India) in terms of greenhouse gas emissions, most of which are accounted for by the fuel and energy sector. Within the Eurasian Economic Union (EAEU), Russia and Kazakhstan account for 97% of all pollutant emissions [48].

EPI and SSI do not contain indicators reflecting the situation with mineral resources as a component of natural capital or indicators for depletion assessment. Only the methodologies for calculating

adjusted net savings (ANS) and the Genuine Progress Indicator (GPI) reflect the cost of mineral resource depletion. However, they do not take into account any indicators characterizing the impact of the natural resources sector in a resource-based country on ecosystems.

None of the indices discussed above reflect the amount of mining waste and its impact on the environment. It is possible that this impact (from both primary and secondary sources of pollution) is included in the indicators reflecting air and water quality but it is not considered as an individual component.

RBEs make the biggest contribution to environmental impact, as a result of which environmental costs are increasing. This has an impact on social well-being, but environmental costs (both investment and operational ones) are not reflected in SD indicators. This means that conclusions cannot be made about the contribution of the mining sector to either pollution or environmental protection.

As has been shown, there is no single universally accepted theoretical and methodological approach to the analysis and assessment of sustainable development. All the indicators which have been developed have their drawbacks.

First, when using systems of issue-specific indicators, the problem arises of selecting indicators that adequately reflect the processes of social, economic, and environmental development, and can be used for their quantitative assessment. Using quantitative indicators based on the SNA can be problematic due to the fact that there are differences between countries' statistical systems and they may present not enough data. Some of the indicators proposed by the UN CSD are mainly focused on assessing the social component of SD, which does not make it possible to analyze the correlation between economic development, its sources, and results in the social and environmental spheres.

Second, when using aggregated indices for comparing or ranking countries, the problem arises of how they or their indicators can be compared. With the development of calculation methods, the number of indicators and the calculation models are changing, which makes it almost impossible to use such indices to reveal trends.

Third, developing qualitative indicators (either issue-specific or aggregated ones) is associated with the use of information that can only be obtained through surveys, which reduces the reliability of the results. Qualitative indicators make it possible to evaluate many social factors, but at the same time, the risk of subjectivity increases (for example, when choosing and substantiating weighting factors that characterize the contribution of various indicators to the total result and can vary). Another problem is how qualitative indicators can be transformed into quantitative ones (for example, in the case of GPI).

Fourth, none of the methods discussed takes into account the specific features of RBEs and their impact on SD, and neither do they take into account the national characteristics of natural resource consumption or the country's place in the international division of labor. RBEs are characterized by the depletion of mineral resources, the accumulation of mining waste, human impact on the environment, and significant investments in environmental protection. This puts resource-based countries at a disadvantage in comparison with countries importing natural capital.

Fifth, aggregated indicators cannot serve as objective criteria for assessing SD in RBEs since they poorly reflect their specific features, including the accumulation of mining waste and human impact on the environment. Also, they do not reflect the value of mineral resources for the economy and the social development of present and future generations, and neither do they cover an aspect such as resource depletion.

Sixth, of all the considered methods for assessing SD, only ANS and GPI contain information on the depletion of mineral resources.

## 5. Conclusions

It has been found that there is no statistically significant correlation between GNI per capita and SSI for the group of countries discussed. Also, there is a moderate negative correlation between GNI per capita and environmental sustainability, a fairly strong positive correlation with economic sustainability, and a weak correlation with social sustainability. The sample is characterized by a strong

correlation between GNI per capita and HDI, which is almost linear. The values of the correlation ratios between GNI per capita and EPI show a high variability with a low degree of correlation. EPI is based on 24 indicators characterizing completely different aspects of environmental health and ecosystem vitality. Such a number of indicators and the fact that changes are frequently made to the methodology result in a rather weak correlation.

An analysis of the SSI and EPI structures showed that they contain some indicators reflecting such aspects as agriculture, biodiversity, climate, water resources and air quality, energy use, water, and energy conservation. Issue-specific indicators reflect negative impact on the environment by categories: heavy metals, air quality and pollution, climate, and energy. In issue-specific indicators, greenhouse gases are separately analyzed as a negative factor affecting the climate.

EPI and SSI do not contain indicators reflecting the situation with mineral resources as a component of natural capital or indicators for depletion assessment. Only the methodologies for calculating adjusted net savings (ANS) and the Genuine Progress Indicator (GPI) reflect the cost of mineral resource depletion. However, they do not take into account any indicators characterizing the impact of the natural resources sector in a resource-based country on ecosystems. It has been revealed that SD indices fail to reflect the specific features of RBEs (such as resource depletion, mining waste accumulation, human impact on the environment, and environmental costs).

None of the indices discussed reflect the complex impact of the mining sector, the amount of mining waste and its impact on the environment, and investments in environmental protection. They only reflect the situation in social, environmental, and economic spheres, which makes it impossible to evaluate the contribution of the mining industry both to pollution and natural resource restoration and means that specific studies are required to analyze these aspects.

The mining sector strongly influences all elements of the natural environment. In terms of impact complexity, this sector ranks first among all other sectors of the economy. The mining sector is a leader in air pollutant emissions and the generation of waste. More than 90% of waste is accounted for by the mining sector, more than half of the waste produced in the mining sector is recycled, and the rest goes to spoil tips or tailings dams, which can act as secondary sources of pollution.

The main environmental expenses were aimed at protecting the atmosphere and water resources. The share of investment in the GDP of the Russian Federation was 0.15–0.16% which is clearly not enough to compared to the level for developed countries, which is1.25–1.5% of GDP. RBEs make the biggest contribution to environmental impact, as a result of which environmental costs are increasing. This has an impact on social well-being.

**Author Contributions:** Conceptualization, T.P. and M.N.; methodology, T.P. and M.N.; software, M.N.; validation, T.P., M.N., and O.M.; formal analysis, M.N. and O.M.; investigation, T.P. and M.N.; resources, O.M.; writing—original draft preparation, T.P., M.N., and O.M.; writing—review and editing, M.N. and O.M.; visualization, M.N. and O.M.; funding acquisition, T.P., M.N., and O.M. All authors have read and agreed to the published version of the manuscript.

**Funding:** This research was funded by RFBR and MCESSM, grant number № 19–510–44013\19.

**Acknowledgments:** Special thanks to Natalia Savelyeva for the administrative and technical support.

**Conflicts of Interest:** The authors declare no conflict of interest involving the results.

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
