# Peer review of "An Assessment of the Applicability of Sustainability Measurement Tools to Resource-Based Economies of the Commonwealth of Independent States"

_sustainability, doi:10.3390/su12145582_

Round 1

Reviewer 1 Report

dear authors, the paper is really interesting.

The subject sounds almost original in the general organisation and in relation with several points of views.

I really appreciated the relationship evaluation between several indexes and sustainabiity indicators.

What looks a little bit strange is the choise of Countries assumed as example for your research. Why Norway (that correctly looks end of scale for many considerations) and not other European Countries? I think it has to be better explained the reason why You choosed these Countries and not others.

Results are a little bit weak in terms of statistical and math correlations. Please, exlain in a better way the statistical foundations of youe research.

Really interesting also the analysis of the environmental impact of the mining sector (based on the statistical data for Russia). But, why you put it in the "discussion" paragraph? I think it represents an important part of your research and I suggest to move it inside the "results". 

Author Response

  1. Your comment is very true. The list of countries was focused on resource-based economies located in Eurasia, but not all of them were studied. To identify the object of the study in a better way, we will reduce the number of countries.

As objects, we select countries that correspond to two conditions at the same time:

1) CIS countries with transition economies:

Source: https://www.imf.org/external/np/exr/ib/2000/110300.htm.

2) countries with economies that can be characterized as resource-based according to the characteristics presented in the article.

Thus, the object of the study is the following six countries: Russia, Kazakhstan, Kyrgyzstan, Uzbekistan, Turkmenistan, and Azerbaijan. (Commonwealth of Independent States)

(1-4)

  1. Thank you for the comment.

It is true that the aim of the study was to establish whether there are statistically significant correlations between GDP in resource-based economies and some sustainable development indicators.

Due to the principal characteristics of resource-based economies, the mining industry accounts for a significant part of GDP. This is why GDP per capita was chosen as the principal indicator.

SSI, EPI, and HDI are the most valid and common indices used for sustainable development assessment in global rankings.

For the purpose of comparability, GDP values were adjusted and presented as normalized values. Normalization was performed for the entire set of countries in the sample.

Then, we get a range of values from ​​0 to 1 on both index and normalized GDP scales. These values ​​characterize the country’s place in the global ranking.

  1. Previously, we used Russia as a case study to analyze the environmental impact of the mining sector. We can completely agree with the recommendation and consider the charts that were plotted as the results of the study. We have made some changes to the objectives of the study.

  1. We agree with you. So the analysis of the environmental impact of the mining sector (based on the statistical data for Russia) have moved inside the "results". (349-450)

Reviewer 2 Report

  • Explain how the considered countries were selected. Ukrain or Finland were not considered while Laos is. 
  • what are "resources-based economies" ? Which resources are you talking about ? energy and non-energy resources ? only fossils or also agriculture ? this should be explained
  • All abbreviations should be explained
  • Extending the linear regression to higher values of GNI leads to HDI>1 and EPI > 100. Does this make sense ? please discuss in more detail the limits of linear regression. Norway is out of the trend defined by lower GNI countries because at least the HDI-GNI evolution is not linear at high GNI (increasing GDP has no effect on HDI that cannot exceed 1). At HDI < 0.9, there is also a strong correlation between  HDI and GDP, so that your results simply reflect the correlation between GNI and GDP, which is not a scoop. A logistic regression including Norway might be better ?
  • The paper often lacks precision. For instance, the GNI is reported in dollars. Current or constant ? If you used current currency, the time evolution of GNI is partly explained by inflation. Which data are plotted In Fig. 3-10 ? for the different countries and different years or for specific years ? The values should be identified by countries and years with different symbols and lines.   
  • The conclusions are very broad and do not really derive from your analysis. It seems that your conclusion is that you used indicators that there are actually useless for the specific case of RBEs.   

Author Response

Answers and comments in the file.

Round 2

Reviewer 2 Report

The authors replied to most of my questions. The procedure of calculation and uncertainties are in some places still difficult to evaluate, but I don't see how the authors could be more informative while keeping the article to a reasonable size. The paper is interesting, the approach is novel and original, the significance of content is high and the ms is worth to be published. 

Author Response

Question: The authors replied to most of my questions. The procedure of calculation and uncertainties are in some places still difficult to evaluate, but I don't see how the authors could be more informative while keeping the article to a reasonable size.

Answer: We have added to the section the methodology of conditions and restrictions on the choice of countries.

(278-284) As objects, we select countries that correspond to two conditions at the same time:

1) CIS countries with transition economies:

Source: https://www.imf.org/external/np/exr/ib/2000/110300.htm.

2) countries with economies that can be characterized as resource-based.

Thus, the object of the study is the following six countries: Russia, Kazakhstan, Kyrgyzstan, Uzbekistan, Turkmenistan, and Azerbaijan.

(339-342) For the purpose of comparability, GDP values were adjusted and presented as normalized values. Normalization was performed for the entire set of countries in the sample. Then, we get a range of values from 0 to 1 on normalized GDP scales. The indexes characterize the country’s place in the global rankings and estimate normalized scoring.
